# Outcome of Neonates Born to SARS-CoV-2-Infected Mothers: Tertiary Care Experience at US–Mexico Border

**DOI:** 10.3390/children9071033

**Published:** 2022-07-12

**Authors:** Colleen Everett, Yae Kye, Sanjeet Panda, Ajay Pratap Singh

**Affiliations:** 1Paul L. Foster School of Medicine, Texas Tech University Health Sciences Center, El Paso, TX 79912, USA; colleen.everett@ttuhsc.edu (C.E.); yae.kye@ttuhsc.edu (Y.K.); sanjeet.panda@ttuhsc.edu (S.P.); 2El Paso Children’s Hospital, 4800 Alberta Ave, El Paso, TX 79905, USA

**Keywords:** neonates, COVID, SARS-CoV-2, pregnant mothers, NICU

## Abstract

Background: SARS-CoV-2 has affected millions of people around the world. There is a need for data on the effects of this infection on neonates admitted to neonatal intensive care (NICU) units born to infected mothers. Here, we decided to analyze neonates born to mothers who tested positive for SARS-CoV-2 and admitted to NICU compared with neonates who remained with their mothers. Methods: All pregnant mothers who tested positive for SARS-CoV-2 during pregnancy between 1 June 2020 and 30 June 2021, along with all neonates born to infected pregnant women, were included in this study. We then compared the neonates admitted to NICU with the neonates who remained with their mothers. Results: Eighty-eight neonates were born to eighty-eight SARS-CoV-2-positive mothers. Fifteen of these neonates were admitted to the NICU. The mothers of the neonates admitted to the NICU were more likely to have received prenatal care outside of the USA. In addition, the neonates admitted to the NICU were more likely to have needed significant resuscitation at birth. Respiratory distress was the most common reason for NICU admission. None of the NICU-admitted neonates were SARS-CoV-2-positive. There were no differences between the values of the complete blood counts, morbidities at discharge, lengths of hospitalization, or rates of readmission to hospital in the first month of life observed between the two groups. Conclusions: The vertical transmission of the SARS-CoV-2 infection remains rare; there was no difference in the hospital outcomes in the neonates of infected mothers. Unlike other studies, which show an increased tendency toward preterm birth in SARS-CoV-2-positive mothers, our study indicates no such association.

## 1. Introduction

Since the first reported case, COVID-19, caused by the SARS-CoV-2 virus, has spread rapidly. On 11 March 2020, the World Health Organization (WHO) declared the COVID-19 outbreak a global pandemic and, to date, the virus has infected over 426 million people, resulting in over 5 million deaths worldwide [1]. In the United States (USA), there have been over 78 million cases and 936,000 deaths [2]. Many studies and investigations have been conducted regarding the viral characteristics, therapies, affected populations, and more. Of these studies, a few specifically examined how SARS-CoV-2 infection in pregnant women affects perinatal and postnatal outcomes.

Pregnancy is a physiological state in which the woman’s body undergoes a series of immunologic adaptations to successfully support another life. It predisposes the mother to increased susceptibility to certain pathogens, including viruses, intracellular bacteria, and parasites [3]. Many reports have investigated the transmission of the SARS-CoV-2 virus in mothers and newborns; vertical transmission seems to be rare [4,5,6,7,8,9,10,11,12]. There is also increasing evidence suggesting that maternal SARS-CoV-2 infection is associated with adverse outcomes, including fetal demise, preeclampsia, preterm birth, and both postpartum and postnatal complications [13,14,15,16]. Given the immunocompromised state of mothers and the increased risk of newborns acquiring viral respiratory infections, understanding the clinical implications and complications, as well as the epidemiological factors, of the SARS-CoV-2 virus is imperative in order to provide the best postpartum and postnatal care [17]. In addition to affecting the vulnerable elderly and immunocompromised populations, the COVID-19 pandemic has disproportionately affected marginalized communities, including ethnic minorities and those of lower socioeconomic status [16,17]. Populations living around the US–Mexico border face unique barriers to obtaining quality care. Studies have shown that disparities in and barriers to access to better health care disproportionately affect people living near border regions [18,19,20].

We aimed to investigate the maternal and postnatal outcomes of SARS-CoV-2-positive mothers admitted at a level IV maternal neonatal care center at US–Mexico border in El Paso, Texas [21]. Our study provides a unique opportunity to examine COVID-19 infections during pregnancy and compare infants admitted to the NICU with those who were not.

## 2. Methods

This study was reviewed by the Institutional Review Board for the protection of human subjects at the Texas Tech University Health Sciences Center in El Paso, Texas, USA, and was approved with a waiver of consent, IRB#079347.

This was a single-center study conducted at an academic level IV maternal and level IV neonatal care center located inside the medical center of Americas at El Paso, Texas.


**Maternal characteristics investigated were as follows:**
Maternal demographics, including prenatal care site and home country.Maternal risk factors.The severity of the COVID-19 infection in pregnant women.Type of treatment for SARS-CoV-2 given to infected pregnant mothers around the time of delivery.



**Neonatal characteristics and outcomes investigated.**


Demographics, including gestation, birth measurements, Apgar scores, and type of neonatal resuscitation needed at birth.Reason for NICU admission, short-term complications and length of hospital stay.Rate of COVID-19 infection immediately after birth and one-month post-hospital discharge.

We analyzed neonates of SARS-CoV-2-infected mothers born between the period of 1 June 2020 and 30 June 2021. We analyzed maternal and neonatal outcomes, comparing infants admitted to the neonatal intensive care unit with those who were not admitted to NICU.

To provide consistency in the type of SARS-CoV-2 testing performed on mothers, we gathered data from 1 June 2020onwards, as, from this time, our Labor and Delivery Unit began using RT-PCR panel testing, which included SARS-CoV-2, exclusively on all pregnant women admitted to the Labor and Delivery Unit. This data collection period also included a time period of about 4 months after the recommendation of two doses of mRNA COVID vaccine was introduced for pregnant women.

In accordance with guidance from AAP and CDC, ICU visits for SARS-CoV-2-infected mothers were restricted until a negative test or health clearance from local health department were provided. In accordance with above guidance, neonates of infected mothers admitted to NICU were isolated in negative-pressure rooms until two negative SARS-CoV-2 PCR tests at 24 and 48 h of age., respectively. Neonates not admitted to NICU stayed with their mothers in their private rooms. Strict contact and droplet precautions were followed by healthcare staff.

All mothers, their newborns, and infants with incomplete data were excluded from the final analysis of this study.

Descriptive statistics (median, 25th, and 75th percentiles for continuous variables; frequencies and percentages for categorical variables) were calculated separately by groups. The chi-squared test or Fisher’s exact test, as deemed appropriate, for categorical variables and the Mann–Whitney test (two-group comparisons) were used to assess statistical significance. A result was considered statistically significant at *p* < 0.05.

## 3. Results

During the study period, from 1 June 2020 to 30 June 2020, a total of 1724 pregnant mothers were tested for SARS-CoV-2 and gave birth. Out of these, a total of 88 (5%) mothers tested positive for SARS-CoV-2 and gave birth (Figure 1). Of these, a total of 15 (17%) neonates were admitted to NICU; the rest 73 (83%) remained with their mothers and did not require NICU admission.

Our institution is located at the border of the United States and Mexico. When comparing the demographics of the pregnant mothers, those who received prenatal care outside the USA were more likely to give birth to infants who required admission to the NICU. We found no difference in maternal age, primary home country, or maternal comorbidities (pre-pregnancy obesity, hypertension, pre-eclampsia, or diabetes) between the two groups. There were no differences in clinical symptoms related to SARS-CoV-2 at the time of positivity, nor in the need or type of treatment that was administered between the two groups (Table 1).

There was no difference when comparing the time of the positive maternal test to the date of delivery (18 ± 43 vs. 17 ± 38 days) between the two groups. There were no differences between the birth gestations of the neonates who were admitted to the NICU and of those who did not need it (39 ± 2 vs. 38 ± 2 weeks). Additionally, there was no difference in birth weight, mode of delivery, sex, or Apgar score between the two groups of neonates (Table 2).

There were two fetal deaths in infected mothers. One was due to natural abortion at 21 weeks, and the second was due to non-resuscitation at 29 weeks of gestation and the choice of comfort-care measures at birth due to fatal meningoencephalocele (Table 2).

We also examined the results of the complete blood count (CBC) tests in both groups and discovered that there was no difference in WBC count, platelet levels, neutrophils percentage, band percentage, or hematocrit percentage between the neonates admitted to the NICU and those who were not (Figure 2).

The neonates admitted to the NICU were more likely to need significant resuscitation at delivery, which included the use of CPAP or endotracheal intubation (60% vs. 2%). Respiratory distress, which included transient tachypnea of the newborn, delayed transition, and retained lung fluid, was the most likely reason for admission to the NICU. None of the infants admitted to the NICU suffered from long-term morbidities (IVH, ROP, and BPD). There was no difference in the median length of hospital stay between the two groups: 3 (2–4) vs. 2 (2–2) days. The infants in both groups had similar rates of breastfeeding upon discharge. There were no differences in the rates of readmission to hospital within the first month of hospital discharge. One baby from the non-NICU group was readmitted to hospital within one month of discharge due to respiratory syncytial virus (RSV)-caused acute respiratory failure and tested negative for SARS-CoV-2 infection. The baby was discharged after a 5-day hospital stay.

## 4. Discussion

To our knowledge, this is the largest detailed retrospective study to analyze the effects of SARS-CoV-2 infection in pregnant mothers and compare neonates admitted to the NICU with those who were not h. Our study confirms the rarity of the vertical transmission of SARS-CoV-2 [22].

Our study indicates that pregnant women who obtained prenatal care outside the USA and in Mexico had a higher number of their infants admitted to NICU. Although we do not know the reason for this outcome, studies have shown that there is evidence of a lower incidence of prenatal care in first-time pregnancies among those of Hispanic ethnicity [23] and in border states in Mexico and the USA [24,25]. We speculate that the combined effects of these risks might have caused the outcomes we observed.

Adult-population studies have shown disproportionate effects of SARS-CoV-2 infection in minority populations [17]. Our study is unique in that the predominant ethnicity of the mothers observed was Hispanic. Previous studies have noted the increased incidence of preterm births in infected mothers [26,27,28,29]. Our study shows no such risk, and we suggest that this may be explained by the fact that the majority of the infected mothers in our study did not suffer from severe symptoms or experience any major COVID-19 complications. It is possible that more severe infection in pregnant women results in higher rates of premature birth and severe neonatal complications [27]. The question of whether there additional genetic or environmental protective effects also play a role needs to be evaluated in in larger studies.

Although none of the neonates admitted to the NICU born to infected mothers tested positive for SARS-CoV-2, they were still more likely to be admitted to NICU with respiratory distress (60% of the infants required either a nasal cannula, CPAP, or mechanical ventilation). The etiology of this observation is unknown. However, studies have shown adverse neonatal effects of exposure to prenatal maternal infections [30], and maternal infection with SARS-CoV-2 has been shown to cause placental inflammation [31]. We suggest that prenatal exposure to SARS-CoV-2 infection could be a possible cause for this abnormal respiratory transition at birth.

Another important observation that emerged from this study is that there was no difference in the length of hospital stay between the neonates admitted to NICU and those who were not. This shows that respiratory distress in these infants is very transient, with quick recovery times, and does not lead to long-term complications.

Studies have shown the absence of the SARS-CoV-2 virus and the presence of SARS-CoV-2 antibodies in breastmilk [32,33,34]. The majority of the infants in both groups who were discharged were fed breastmilk. Interestingly, we found no evidence of infected mothers transmitting SARS-CoV-2 infections to their neonates that were severe enough to cause readmission to the hospital within one month of discharge. We attributed this observation to the provision of passive immunity through the protective properties and antibodies present in breastmilk, preventing the development of severe COVID-19 disease in the infants of the infected mothers [35]. This outcome solidifies the recommendation that infected mothers to continue provide breastmilk to their neonates [36,37].

Due to uncertainties about the effects of this novel coronavirus infection, very early during the pandemic, our institution introduced a policy of obtaining complete blood counts at 24 and 48 h of age in all infants born to SARS-CoV-2-infected pregnant mothers. This gave us an opportunity to evaluate the laboratory results of these neonates of infected mothers. We found no differences in routine inflammatory markers of infection, such as white blood cell count, neutrophil percentages, or band percentage between the neonates admitted to NICU and those who were not. Our study shows that the performance of routine laboratory tests on all newborns and, especially infants from SARS-CoV-2-positive mothers, does not change outcomes and should be discouraged [38].

Our study has several limitations. First, this is a retrospective study, and we can only ascertain associations and not causality. Second, we acknowledge that our sample size is small, and a larger study is needed to confirm these results. Third, incomplete data about the complete blood counts were used for all the neonates. Fourth, due to the mobility of people across the border, post-natal follow-up may have been lost, resulting in the possible undercounting of the post-discharge data of the infants; however, our institution is the only free-standing children’s hospital in the region; consequently, the majority of local children usually seek care at our institution. Fifth, our study did not examine the placental pathologies of the infected pregnant mothers, which could have shed more light on the reasons for some of the outcomes. A larger study comparing the infants of SARS-CoV-2-infected pregnant mother with those of non-infected mothers at the US–Mexico border may provide more insight. Another aspect that could affect results of similar studies in the future is better acceptance of COVID-19 vaccines by pregnant mothers, as none of the pregnant mothers in this study had received any of their COVID-19 vaccines, even when these were widely available.

## 5. Conclusions

The coronavirus pandemic has had and continues to have a dramatic effect on the lives of all humans. Fortunately, vertical transmission from infected pregnant mothers remains rare. In this study of a predominantly Hispanic population, maternal SARS-CoV-2 infection during pregnancy resulted in a greater need for resuscitation in neonates admitted to NICU, but did not result in a higher incidence of preterm birth. The infected mothers we less likely to transmit infection severe enough to cause hospitalization within one month of hospital discharge. This study also solidifies the recommendation that infected mothers continue to provide breastmilk

## Figures and Tables

**Figure 1 children-09-01033-f001:**
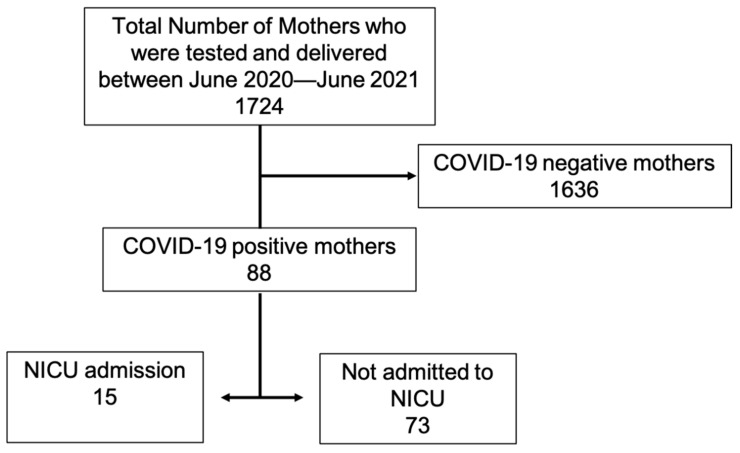
Distribution of patients.

**Figure 2 children-09-01033-f002:**
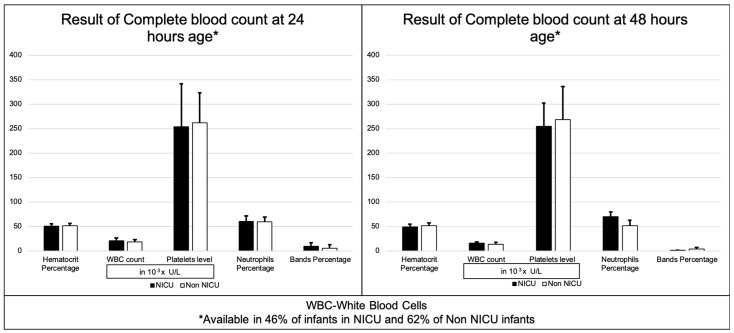
Results of complete blood count at 24 and 48 h.

**Table 1 children-09-01033-t001:** Maternal characteristics of SARS-CoV-2-infected mothers of neonates admitted to the NICU and those who were not.

Parameters	COVID-Positive NICU Mothers 15	COVID-Positive Non-NICU Mothers 73	*p*-Value
Maternal Age In years	26 ±6	27 ± 6	NS
Ethnicity Hispanic	87%	96%	NS
* Primary home country Mexico USA	3 (20%) 12 (80%)	16 (22%) 56 (78%)	NS
^$^ Primary prenatal care country Mexico USA	9 (60%) 6 (40%)	18 (25%) 53 (75%)	0.02
Maternal co-morbidities None Obese Diabetes Gestational hypertension Pre-eclampsia Asthma Illicit drug abuse	46% 23% 15% 15% 8% 8% 0%	44% 30% 10% 10% 13% 3% 3%	NS
Asymptomatic at the time of testing No symptoms	85%	78%	NS
Treatment given specifically to COVID Symptomatic ^&^ or none Face-mask oxygen Steroids Others	100%	93% 1.5% 1.5% 4%	NS

* Primary home country was defined using the given address of the contact at the time of admission. ^$^ Primary prenatal-care country was defined as receiving fewer than three visits at any clinic/hospital in the USA. ^&^ Symptomatic—antipyretics and painkillers. NS—non-significant.

**Table 2 children-09-01033-t002:** Characteristics of neonates of SARS-CoV-2-positive mothers admitted to the NICU vs. those who were not.

Neonatal Parameters	COVID-Positive NICU	COVID-Positive Non-NICU	*p*-Value
COVIDpositive mothers	15	73	
Days to delivery from testing positive	18 ±43	17 ±38	NS ^Ω^
Median gestation at maternal positive test	39 (37–40)	38 (36–39)	NS
COVID-positive baby	0 (0%)	3 (4%)	NS
Birth gestation weeks ^a^	39 ± 2	38 ± 2	NS ^Ω^
Birth weight (grams) ^a^	3271 ± 435	3176 ± 577	NS ^Ω^
Birth head circumference (cm) ^a^	34 ± 1	34 ± 3	NS ^Ω^
Birth length (cm) ^a^	50 ± 1	50 ± 3	NS ^Ω^
Mode of delivery C-section Vaginal	33% 67%	21% 79%	NS
Sex Male Female	66% 34%	50% 50%	NS
Apgar ^a^ 1 min 5 min	8 ±1 9 ±1	8 ±1 9 ±1	NS ^Ω^
Delayed cord clamping	15%	38%	NS
Highest resuscitation at delivery None Nasal cannula or CPAP Endotracheal intubation	40% 47% 13%	98% 2%	0.0001
Reason for NICU admission Respiratory distress ^$^ Suspected sepsis due to chorioamnionitis Prematurity Other	46% 33% 6% 15%	Not applicable	
Neonatal morbidity Any (IVH, BPD, NEC, other) ^%^	0%	0%	NS
Deaths	0	2 ^@^	NS
Length of hospitalization Median (quartile)	3 (2–4)	2 (2–2)	NS
Feed type at discharge— breastmilk	87%	82%	NS
Readmission to hospital in 1st month	0%	1 baby with COVID-19 respiratory distress	NS

^%^ IVH—any intraventricular hemorrhage, BPD—bronchopulmonary dysplasia defined per NICHD consensus statement as need for oxygen support at 36 weeks of postmenstrual age (PMA), NEC—any necrotizing enterocolitis. ^$^ Respiratory distress—transient tachypnea of newborn, delayed transition, or retained lung fluid. @ 1 due to abortion at 21 weeks and 1 due to meningoencephalocele at 28 weeks and comfort care. a—mean ± SD; Ω—Student’s unpaired t-test; remainder by Fisher’s exact test. NS–not significant.

## Data Availability

The data presented in this study are available on request from the corresponding author.

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
