# Peer review of "Outcome of Neonates Born to SARS-CoV-2-Infected Mothers: Tertiary Care Experience at US–Mexico Border"

_children, 2022, doi:10.3390/children9071033_

Round 1

Reviewer 1 Report

Dear authors,

I think it would be interesting to analyze the motive of NIUC admission of babies from non-infected mothers (the manuscript did not present this data) and to compare it with the 88 cases of positive mothers. You analyze the characteristics of neonates of SARS-Cov-2 positive mothers admitted to the NICU vs those who were not admitted. This is the biggest flaw in this analysis, the comparison shouldn’t be between admitted vs. not admitted to the NICU but realizing if there is anything different between those born to positive mothers and non-infected mothers. Also, the hematological markers must be compared with babies from non-infected mothers (or reference). It would be worthwhile to analyze the reasons for NICU admission compared to previous years before pandemics (epidemiological data). Health services were under great pressure and probably care protocols for pregnant women have inevitably changed given the COVID-19 emergency. 

Author Response

I think it would be interesting to analyze the motive of NIUC admission of babies from non-infected mothers (the manuscript did not present this data) and to compare it with the 88 cases of positive mothers. –

  • We agree with the comment made by the reviewer here, that a study designed to compare babies of covid infected mothers vs babies on non-infected mothers would be interesting, however our goal here was analyze outcome of newborns of COVID infected mothers only. We wanted to analyze maternal characteristics, reasons for NICU admissions, short morbidities, and out of hospitalization outcome of infants, to tease out differences in neonates admitted to NICU to shed light on risk factors which put infants of infected mothers at higher risk of nicu admission.

You analyze the characteristics of neonates of SARS-Cov-2 positive mothers admitted to the NICU vs those who were not admitted. This is the biggest flaw in this analysis, the comparison shouldn’t be between admitted vs. not admitted to the NICU but realizing if there is anything different between those born to positive mothers and non-infected mothers.

  • Here again, we agree with the reviewer a study designed to analyze infants of infected vs non infected mothers would be important. Such, large studies have already been published. We wanted to only analyze and characterize infants of only infected mothers. We decided upon this objected due to our unique population at the US-Mexico border. We have made a following addition to the discussion part to better reflect this.

“A larger study comparing infants of SARS-CoV-2 infected pregnant mother with those of non-infected at the US-Mexico may provide more insight”

Line 214-215

Also, the hematological markers must be compared with babies from non-infected mothers (or reference). It would be worthwhile to analyze the reasons for NICU admission compared to previous years before pandemics (epidemiological data). Health services were under great pressure and probably care protocols for pregnant women have inevitably changed given the COVID-19 emergency. 

  • We agree with the reviewer here comparing hematological markers between infected and non-infected mothers would have been interesting. We could not compare previous data as the objective of this study was only to look at outcome of neonates of infected mothers. We also agree that health services were under tremendous pressure at the peak of this pandemic. Fortunately, we did not have too many infected pregnant mothers, only protocol that we had to change for NICU was infected mothers had to wait until a negative test or after given a clearance from local health department to be allowed to come and visit their babies in NICU. No such restrictions were placed in well baby nursery for those not admitted to NICU.

We have added following statement to methods and discussion

“In accordance with guidance from AAP and CDC visitation for SARS-Cov-2 infected mothers to NICU was restricted until a negative test or health clearance from local health department”

“In accordance with above guidance neonates of infected mothers admitted to NICU were isolated in negative pressure rooms until two negative SARS-CoV-2 PCR tests at 24 and 48 hours of age respectively. Neonates not admitted to NICU stayed with their mothers in their private rooms. Strict contact and droplet precautions were followed by healthcare staff.”

Line 93-100

Reviewer 2 Report

- Please review capitalization and punctuation.

- Point out that these data were gathered before vaccines had become available.

Line 18 - You mean any time during the pregnancy from first trimester to time of delivery?

Line 72 - I'm not sure if I would really call these information "outcomes"

Line 78 - Briefly discuss hospital policy if any, on rooming in/ breastfeeding babies born to COVID + moms

Line 116-117 Table 1- Include in footnote what NS stands for; either you report % symptomatic or % asymptomatic. Having a heading symptomatic and reporting data for no symptoms looks confusing; also specify/ enumerate what Others under COVID treatment were

Line 133 - In case, baby had multiple CBC's which CBC are you using for the data?

Line 134-135 Table 2 - COVID (+) baby - discuss how they were managed and mom's age of gestation when she tested (+) for COVID; Prematurity under reason for NICU admission -- discuss how premature these babies were

Line 151- Can you talk more about that 1 baby that got readmitted?

Line 154 - Great discussion

Line 220-221 - I would shy away from making this generalization since we didn't really compare these events with non-CoVID births

Author Response

Reviewer 2 comments

Please review capitalization and punctuation.

  • Yes, we have reviewed that and made changes to reflect them. Thank you.

- Point out that these data were gathered before vaccines had become available.

  • This data is from within the time period when mRNA vaccines became available and were just started to being recommended for pregnant women. We have reviewed our data set again and found that none of pregnant women in this study received any Covid-19 vaccines.

We have added following statement to methods.

“This data collection period also included time period of about 4 months after two of mRNA covid vaccine were started to being recommended for pregnant women.”

We have added the following statement to the discussion

“Another aspect which could affect results of similar studies in future is better acceptance of covid-19 vaccines by pregnant mothers as none of the pregnant mothers in this study received any of the covid-19 vaccines even after wide availability”

Line 18 - You mean any time during the pregnancy from first trimester to time of delivery?

  • Yes, any time during pregnancy or upon being tested at the time of admission to labor and delivery.

Line 72 - I'm not sure if I would really call these information "outcomes"

  • Thank you for pointing this out. We have changed this to the following.

“Maternal characteristics investigated”

“Neonatal characteristics and outcomes investigated”

Line 78 - Briefly discuss hospital policy if any, on rooming in/ breastfeeding babies born to COVID + moms

  • Just like with all NICU across the world visitation policy did change after onset of pandemic. We have added the following sentence in our methods.

“In accordance with guidance from AAP and CDC visitation from SARS-Cov-2 infected mothers to NICU was restricted until a negative test or health clearance from local health department”

Line 116-117 Table 1- Include in footnote what NS stands for; either you report % symptomatic or % asymptomatic. Having a heading symptomatic and reporting data for no symptoms looks confusing; also specify/ enumerate what Others under COVID treatment were

  • Thank you for this suggestion.
  • We have added the following to table 1

“NS- Non significant”.

We changed to “Asymptomatic at the time of testing”

Line 133 - In case, baby had multiple CBC's which CBC are you using for the data?

  • We only analyzed CBC’s that were drawn as close to the age 24 hours and 48 hours as possible to keep consistency. Hence, figure 2 shows only 46% and 62% of labs availability at this age.

Line 134-135 Table 2 - COVID (+) baby - discuss how they were managed

  • Thank you for this comment.
  • There was no difference in management protocol for neonates of infected mother except for contact and droplet precautions along with staying in negative pressure isolation room until neonate’s SARS-CoV-2 PCR test resulted negative at 24 and 48 hours age.

We have added the following statement to methods section here.

“In accordance with interim AAP guidance neonates of infected mothers admitted to NICU were isolated in negative pressure rooms until two negative SARS-CoV-2 PCR tests at 24 and 48 hours of age respectively. Neonates not admitted to NICU stayed with their mothers in their private rooms. Strict contact and droplet precautions were followed by healthcare staff.”

and mom's age of gestation when she tested (+) for COVID;

  • We have indicated this the showcasing days to delivery from the time of positive test in table 2
  • But we have also added Gestation at the time of positive covid test as suggested by the reviewer in Table 2

Prematurity under reason for NICU admission -- discuss how premature these babies were

  • There was only one baby admitted to NICU who was premature at 34 weeks gestation.

Line 151- Can you talk more about that 1 baby that got readmitted?

  • This child was admitted on day of life 17 with RSV induced acute respiratory failure and was COVID-19 negative during this admission. Thank you for this keen observation.

We have added the following to our results

“One baby from non-Nicu group was readmitted to hospital within one month post discharge due to RSV caused acute respiratory failure and tested negative for SARS-CoV-2 infection”

Line 154 - Great discussion

  • Thank you

Line 220-221 - I would shy away from making this generalization since we didn't really compare these events with non-CoVID births

  • We have amended the lines.

“In this study with Hispanic predominant population, the maternal SARS-CoV-2 infection during pregnancy resulted in higher need of resuscitation in neonates admitted to NICU but did not result in a higher incidence of preterm birth. Infected mothers we less likely to transmit infection severe enough to cause hospitalization within one month post hospital discharge.”

Round 2

Reviewer 1 Report

Dear authors,

Dear editors

I was satisfied with your answers.

The manuscript is improved and I think it has relevant information. The characteristics of neonates of SARS-Cov-2 positive mothers admitted to the NICU vs those who were not and maternal characteristics of SARS-CoV-2 infected mothers are relevant for future clinical management.